

# Removing spurious inertial instability signals from gravity wave temperature perturbations using spectral filtering methods

Cornelia Strube[1], Manfred Ern[1], Peter Preusse[1], and Martin Riese[1]

[1]Institut für Energie- und Klimaforschung – Stratosphäre (IEK–7), Forschungszentrum Jülich GmbH, 52425 Jülich, Germany

**Correspondence:** Cornelia Strube (c.strube@fz-juelich.de)

**Abstract.** Gravity waves are important drivers of dynamic processes in particular in the middle atmosphere. To analyse atmospheric data for gravity wave signals it is essential to separate gravity wave perturbations from atmospheric variability due to other dynamic processes. Common methods to separate small-scale gravity wave signals from a large-scale background are separation methods depending on filters in either the horizontal or vertical wavelength domain. However, gravity waves are

not the only process that could lead to small-scale perturbations in the atmosphere. Recently, concerns have been raised that vertical wavelengths filtering can lead to misinterpretation of other wave-like perturbations, such as inertial instability effects, as gravity wave perturbations.

In this paper we assess the ability of different spectral background removal approaches to separate gravity waves and inertial instabilities using artificial inertial instability perturbations, global model data and satellite observations. We investigate a

horizontal background removal, which applies a zonal wavenumber filter with additional smoothing of the spectral components in meridional and vertical direction, a sophisticated filter based on 2D time-longitude spectral analysis (see Ern et al. (2011)) and a vertical wavelength Butterworth filter.

Critical thresholds for the vertical wavelength and zonal wavenumber are analysed, respectively. Vertical filtering has to cut deep into the gravity wave spectrum in order to remove inertial instability remnants from the perturbations (down to 6 km

cutoff wavelength). Horizontal filtering, however, removes inertial instability remnants in global model data at wavenumbers far lower than the typical gravity wave scales for the case we investigated. Specifically, a cutoff zonal wavenumber of 6 in the stratosphere is sufficient to eliminate inertial instability structures. Furthermore, we show that for infrared limb-sounding satellite profiles, it is possible as well to effectively separate perturbations of inertial instabilities from those of gravity waves using a cutoff zonal wavenumber of 6. We generalize the findings of our case study by examining a one-year time series of

SABER data.

## 1 Introduction

In the middle atmosphere, various wave and wave-like processes shape the global structures of temperature and winds. This includes planetary waves (Rossby waves) (Salby, 1984), tropical wave modes (e.g. Kelvin waves) (Ern et al., 2008; Baldwin et al., 2001), inertial instabilities (Knox and Harvey, 2005) and gravity waves (Fritts and Alexander, 2003; Alexander et al.,

2010). These processes usually exhibit very different scales in at least one of the spatial dimensions. For the evaluation of



measurement data from ground-based, in-situ and satellite instruments, a spatial or temporal scale separation of waves and wave-like processes in the atmosphere is key to analysing the effects that one specific process has on the atmosphere. Accordingly, most analysis approaches for gravity wave activity rely on an initial "background removal" using spatial scale separation to extract only the signal introduced by gravity waves. How efficient different background removals are, is highly dependent on the method used and on the kind of other perturbations present.

Recently, Rapp et al. (2018a) derived a global, gravity wave temperature perturbation climatology from MetOp GPS radio occultation (GPS-RO) data and found unexpectedly high potential energy densities in the tropics that could not be linked to gravity wave sources. In further investigations, they were able to tie the unexpectedly high gravity wave activity to inertial instability signals remaining in their gravity wave perturbations (Rapp et al., 2018b). They showed the existence of a perturbation structure in their measurements with large horizontal and small vertical extent in the region of interest and demonstrated with a vorticity argument that inertial instability induced those additional perturbations.

Inertial instabilities in a rotating-stratified fluid, like the Earth's atmosphere, arise from an imbalance between the pressure gradient and the centrifugal forces when the absolute angular momentum decreases with the radius (Dunkerton, 1981). To stabilize the imbalance, horizontal circulation generates transport of angular momentum in meridional direction. This results in shallow vertical motions accompanying the meridional flow (Rapp et al., 2018b). This flow causes layered temperature perturbations observed around the equator in the stratosphere (Hitchman et al., 1987; Hayashi et al., 1998; Smith and Riese, 1999). Flow distortion by planetary waves can trigger zonal variations in winter and thereby create local instability that extends into the midlatitudes (Smith and Riese, 1999). Stratospheric inertial instability signals are commonly found at the equatorial stratopause and along channels extending to the winter hemisphere midlatitudes and down into the middle stratosphere (Knox and Harvey, 2005). The analysis of Rapp et al. (2018a) has shows the frequent occurrence of midlatitude inertial instability signals around the northern winter solstice.

Gravity waves are small-scale oscillations in the atmosphere that are balanced by buoyancy as restoring force. They are generated mostly in the troposphere by flow over orography, imbalances along jet streams and frontal systems and convection e.g. over storm systems (Fritts and Alexander, 2003; Alexander et al., 2010). Breaking and dissipating gravity waves exert drag on the atmospheric circulation. The energy and momentum carried upward by gravity waves contribute to the stratospheric and mesospheric branches of the Brewer-Dobson circulation (Alexander and Rosenlof, 2003; McLandress and Shepherd, 2009; Butchart et al., 2010), the quasi-biennial oscillation in the stratospheric winds (Dunkerton, 1997; Baldwin et al., 2001; Ern et al., 2009, 2014) and the semiannual oscillation at the stratopause (Garcia et al., 1997; Ern et al., 2015). Furthermore, gravity waves contribute in setting the conditions for sudden stratospheric warmings (Wright et al., 2010; Yamashita et al., 2010; Albers and Birner, 2014; Šácha et al., 2016; Ern et al., 2016).

In the last decades, satellite measurements began to allow insights into global distributions of gravity wave activity (Alexander et al., 2010; Geller et al., 2013). Since the early approaches of Fetzer and Gille (1994); Wu and Waters (1996) and Eckermann and Preusse (1999) many studies have been initiated to collect gravity wave information from space using techniques like limb sounding (Preusse et al., 2002; Alexander et al., 2008a; Wright and Gille, 2011; Ern et al., 2018), microwave sub-limb sounding (McLandress et al., 2000; Jiang et al., 2004b, a; Wu and Eckermann, 2008), infrared and microwave nadir sounding





(Wu, 2004; Eckermann et al., 2006, 2007; Hoffmann and Alexander, 2009; Gong et al., 2012; Hoffmann et al., 2016; Wright et al., 2017; Ern et al., 2017) and GPS-RO (Tsuda et al., 2000; Fröhlich et al., 2007; Schmidt et al., 2008; Wang and Alexander, 2010; Šácha et al., 2016; Rapp et al., 2018a). All these measurements rely on scale separation to discern gravity waves signals from a background atmosphere.

Historically, different background removal techniques have been applied for different measurement methods because of their specific characteristics (Alexander et al., 2010). For instance, vertical high-pass filters have often been used for GPS-RO measurements because the observed temperature profiles are quasi-randomly scattered over the globe. On the other hand, horizontal low-pass filters have been applied to data from IR spectrometry sounders due to their regular coverage of the whole earth. The different background removal techniques also influence the spectrum of waves visibly in the residual data and thus

the background removal is an important part of the observational filter.

Alexander (1998) first discussed the influence of observational filters from different measurement systems, which may largely alter the apparent global distributions (Preusse et al., 2001). This implicitly includes the influence of different background removal techniques. John and Kumar (2012) then compared the influence of using different background removal methods for the same data set. They found that the difference introduced by the application of vertical instead of horizontal filtering

exceeds the differences between data sets applying the same method significantly. Schmidt et al. (2016) confirmed that different techniques (in particular vertical spectral filtering versus horizontal S-transform analysis) have a large impact on the global gravity wave activity results, especially comparing gravity wave potential energy densities and gravity wave momentum flux estimates. In addition, they strongly recommend to use horizontal filtering for the detrending: On the one hand, vertical filtering underestimates gravity wave activity in the polar vortex region, where high wind speeds refract waves to larger ver-

tical wavelengths. On the other hand, vertical detrending may overestimate gravity wave activity due to remnant signals from synoptic and planetary waves and their small vertical scale in the tropics. Inertial instabilities have not been considered in these discussions.

Rapp et al. (2018b) further showed in their study that subtracting a large-scale background estimated from ERA-Interim reanalysis temperatures can eliminate spurious inertial instability signals from the gravity wave temperature perturbations and

leaves gravity wave activity around the polar vortex. In a comment on this study, Harvey and Knox (2019) pointed out that it is not clear how often inertial instabilities induced variations have been misinterpreted as gravity wave signals in the past. They argue that the signals from inertial instabilities, if present, have to be identified consciously by investigating the large-scale meteorologic situation in areas prone to inertial instability.

In this study, we investigate the application of both a vertical and a horizontal gravity wave background removal on different

temperature data sets that incorporate inertial instability signals and evaluate to what extent the background removals are able to remove these instability signals from the gravity wave perturbations. In particular, we evaluate what cutoff length scales are optimal as a compromise between preserving the full gravity wave spectrum and removing more remnant structures due to inertial instabilities. For a first access to the topic, we start by analysing artificial inertial instability perturbations. The structure was modeled according to the conditions found for a strong inertial instability event in December 2015, as previously discussed

in Rapp et al. (2018b). Then, we focus on a vertical and horizontal filtering approach on ERA5 temperatures. Accordingly,



the methodic limitations of vertical and horizontal scale separation approaches can be tested showcasing the challenge of removing a background in a more complex atmospheric data set, where inertial instability signals, gravity wave perturbations and other fluctuations are simultaneously present. Furthermore, we investigate the sensitivity of the methods on the removed characteristic length scales to estimate an optimal cutoff parameter in the particular case. Lastly, we evaluate the performance of

a dedicated SABER (Sounding of the Atmosphere using Broadband Emission Radiometry) background removal, as introduced by Ern et al. (2011, 2018), in time and longitude dimensions in comparison to the Butterworth filter applied by Rapp et al. (2018a, b) with respect to spurious inertial instability signals and the reliability of the background removals for the December 2015 case as well as in the course of one year of SABER data.

This paper is structured as follows: Section 2 describes the different data sets we use in the study. Section 3 introduces the

background removal techniques and measures for evaluating the resulting perturbations. In Sect. 4, we apply these techniques to the data and assess their performance, and Sect. 5 summarizes our findings.

## 2 Data

For different steps of our analysis, we utilize the characteristics of satellite observations, realistic atmospheric reanalysis data and an idealized data set that only contains inertial instability perturbations. Satellite measurements give unique information

by providing global sampling on real observations. The reanalysis data set has the advantage of covering a large range of spatial and spectral scales without being restricted to a specific, irregular sampling. The artificial perturbation model gives the opportunity to see how the background removal techniques are influencing an isolated inertial-instability-like signal. Any residual fluctuations after applying the background removal then indicate remnant inertial instability signals that should not be attributed to other processes. In realistic model data sets or measurement results possibly remnants signals might belong to the

gravity wave activity we eventually want to infer. This section describes the various data sets.

### 2.1 SABER temperatures

The Sounding of the Atmosphere using Broadband Emission Radiometry (SABER) instrument is operated onboard the Thermosphere Ionosphere Mesosphere Energetics and Dynamics (TIMED) satellite, which was launched in December 2001 and is still operational to this date. The limb-sounding instrument records profiles of the atmospheric radiation by scanning its viewing

direction vertically and covers a large altitude range from the tropopause to well above 100 km. For this study, we are mainly interested in the stratosphere and hence consider altitudes $\leq 60$ km. The tangent point track, i.e. the track of observation points, is roughly 2000 km away from the orbital track. Temperatures are predominantly retrieved from the 15 μm $CO_2$ infrared band. Retrieval results are provided in vertical profiles with about 2 km vertical resolution, 400 m vertical sampling and spaced on average 400 km along the tangent-point track. Detailed information on the retrieval are summarized in Remsberg et al. (2008).

The TIMED satellite covers almost the whole globe with ∼15 orbits/day. Hence, waves up to a zonal wavenumber of 6 to 7 can be resolved along a latitude circle (Salby, 1982). Furthermore, the satellite performs yaw maneuvers close to every 60 days. This alternates the SABER viewing geometry between 50°S to 82°N, northward viewing, and 82°S to 50°N, southward





viewing. While latitudes equatorward of 50° are covered continuously, the higher latitudes are only covered 60 days in every 120 days.

According to Remsberg et al. (2008), the precision, i.e. the random instrument error, for the SABER temperature measurements in the V1.07 retrieval results to 0.3 to 0.6 K (mean at 0.45K) in the stratosphere (between 20 and 50 km altitude). The precision error is predominantly related to pointing jitter. In this study we use temperatures obtained with the V2.0 retrieval algorithm. Ern et al. (2018) and the official SABER website (http://saber.gats-inc.com/temp_errors.php) report that there are no changes in the retrieval between V1.07 and V2.0 that would impact the precision of the temperature estimates, but the overall

accuracy could be altered.

## 2.2    ERA5 temperatures

ERA5 is the newest reanalysis product based on the 4D-Var data assimilation of ECMWF's Integrated Forecast System (IFS). In the simulation, the grid consists of 137 hybrid model levels with a top at 0.01hPa. In the stratosphere, the hybrid levels are less than 1km apart. The IFS is a spectral system. In version cy41r2, the model has a horizontal resolution of $\sim$0.28 ° or $\sim$31 km.

Power spectral analysis showed that only gravity waves with wavelengths larger than a few grid points ($6\Delta x$ ; in this case $\sim 1.7°$) may be expected to be well resolved in the model output (Jewtoukoff et al., 2015). Recently, ERA5 has been shown to represent the stratospheric temperature structure with more details than other reanalysis products (Wright and Hindley, 2018). For our analysis, we use a global, interpolated field regularly sampled to $0.2° \times 0.2° \times 0.5$ km (longitude $\times$ latitude $\times$ altitude) resolution. Hence, we expect an effective resolution of $\sim 180$ km at the equator or $\sim 90$ km at 60° latitude. This covers a large

range of the gravity wave spectrum as well as inertial instability signals.

## 2.3    Artificial inertial instability perturbations

We construct a simple model of artificial inertial instability temperature perturbations ($T'_{II}$) to first evaluate the ability of the background removal on isolated signals. In building the model, we consider both the theoretical knowledge about the spatial structure of stratospheric inertial instability effects (Dunkerton, 1981; Knox and Harvey, 2005) and the situation observed in

the particular inertial instability case on 3 December 2015. For each location $\boldsymbol{x} = (x, y, z)$, the perturbation value is derived from a constant temperature perturbation $T'_{II}$ and three independent spatial shape functions $w_*$ to form the structure.

$$T_{II}(x, y, z) = T'_{II} \cdot w_{zonal}(x) \cdot w_{meridional}(y) \cdot w_{vertical}(z). \tag{1}$$

Following the findings from Dunkerton (1981), we use the first derivative of a Gaussian as meridional component.

$$w_{meridional}(y) = -\frac{y - y_0}{\beta} \exp\left(-\frac{1}{2}\left(\frac{y - y_0}{\beta}\right)^2\right) \tag{2}$$

The symmetric structure is centered around the latitude $y_0$ and defined by a characteristic width $\beta$.





**Table 1.** Parameters adapted for this study in the artificial inertial instability perturbation model defined in Eqs. 1 to 4.

| $T'_{II}$ | $y_0$ | $\beta$ | $x_1$ | $x_2$ | $\alpha$ | $z_0$ | $m$ |
|-----------|-------|---------|-------|-------|----------|-------|-----|
| [K] | [°] | [°] | [°] | [°] | [°] | [km] | [km$^{-1}$] |
| 12 | 20 | 15 | 240 | 40 | 40 | 2 | $\frac{2\pi}{12}$ |

In the vertical, we apply a sinusoid with a given vertical wavenumber $m$ to fit the wave-like "pancake" structure. The lowest level with zero perturbation is defined by $z_0$.

$$w_{vertical}(z) = \sin(m(z - z0)) \tag{3}$$

For the zonal dimension, we use a step function with smoothed transition to zero in order to restrict the structure to about a third of a latitude circle, since midlatitude inertial instability signals are often constrained in a so-called "channel" (Knox and Harvey, 2005). To define the step function, we use the edge longitudes $x_1$ and $x_2$, that enclose the longitude range containing the nonzero perturbations, and a width $\alpha$ for the smooth transition from weights 0 to 1. To simplify the notation, we use $\alpha_1 := x_1 + \alpha$ and $\alpha_2 := x_2 - \alpha$ as auxiliary parameters.

$$w_{zonal}(x) = \begin{cases} 1 & x \in [\alpha_1, \alpha_2] \\ -\frac{1}{2}\cos\left(\pi\frac{x}{\alpha}\right) + \frac{1}{2} & x < \alpha_1, x \in [x_1, x_2] \\ -\frac{1}{2}\cos\left(\pi\frac{\alpha_2 - x}{\alpha}\right) + \frac{1}{2} & x > \alpha_2, x \in [x_1, x_2] \\ 0 & x \notin [x_1, x_2] \end{cases} \tag{4}$$

In order to constrain the free parameters of the three shape functions, we compare to the case of 3 December 2015. Based on a one-year survey of GPS-RO temperature anomalies and a climatological analysis of ERA-Interim data by Rapp et al. (2018b), this case of 3 December 2015 is a particular strong one. The vertical and horizontal scales of the inertial instability are representative for typical winter conditions. The same inertial instability signals can also be found in ERA5 and SABER temperatures, as will be discussed in more detail in Section 4. Figure 1 shows vertical, longitudinal and latitunal cross sections as well as the zonal mean structure in a restricted longitude band (similar to Fig. 1b) in (Rapp et al., 2018b)). The upper row depicts the structures of the artificial data set based on Eqs. 1 to 4 and the parameters from Table 1 to generate the structure illustrated. The lower row gives the corresponding plots from ERA5 temperature perturbations, calculated using vertical filtering with a fifth order Butterworth filter with a cutoff vertical wavelength of 15 km. (The Butterworth filter will be explained in more detail in Sect. 3.1.) The lower row also shows the modeled inertial instability signal as white contours for comparison with the ERA5 structures.



## 3 Background removal methods and diagnostic quantities

In the analysis, we have to distinguish between "regularly sampled" such as outputs from an atmospheric model and "sparsely sampled" such as observations by satellite instruments, though both kinds of data sets are global. This section describes the various methods for background removal applicable to these different data sets.

### 3.1 Vertical filtering

As a first approach of a background removal, we apply is a Butterworth filter defined by a window function $H(\lambda_z)$

$$H(\lambda_z) = \left(1 + \left(\frac{\lambda_z}{\lambda_c}\right)^{2n}\right)^{-\frac{1}{2}} \tag{5}$$

with the cutoff wavelength $\lambda_c$. The vertical wavelength is defined as $\lambda_z = \frac{2\pi}{m}$ and $m$ represents the vertical wavenumber. The order of the Butterworth window is given by $n$.

The Butterworth filter was used by Rapp et al. (2018a, b) in their analysis of MetOp-GPS temperature profiles and before introduced in detail by Ehard et al. (2015). Vertical filtering has the advantage that it can be applied profile-wise without assuming any correlation between (adjacent) profiles. It is therefore particularly suited for e.g. GPS-RO and ground-based lidar measurements (Rapp et al., 2018a), but can also be employed easily on model products like ERA5 or the artificial inertial instability signals, which we built in Sect. 2. The Butterworth filter is a high-pass Fast Fourier Transform (FFT) filter, i.e. we define the gravity wave perturbation as the inverse transform from Fourier-space, where scales longer than the critical vertical wavelength $\lambda_c$ were smoothly removed by convolving the Butterworth window.

Where we refer to vertical filtering in this paper, we usually use $n = 5$ and $\lambda_c = 15$km as previously applied in Ehard et al. (2015) and Rapp et al. (2018b). The exception is the analysis presented in Section 4.3, where we assess the inertial instability removal for different $\lambda_c$ values.

### 3.2 Horizontal filtering for global, regularly sampled data sets

For horizontal filtering in regularly sampled, global snapshots, we apply zonal FFT in longitude direction to provide a zonal wavenumber spectrum for each altitude and latitude. The background is defined as the lowpass up to a cut-off wavenumber, $k_c$, i.e. we set the spectral components corresponding to wavenumbers larger than $k_c$ to zero. If not stated differently, we apply a cutoff zonal wavenumber $k_c = 6$ in the stratosphere. This cutoff has been used in a large number of previous studies on low earth orbiter (LEO); (e.g. Fetzer and Gille, 1994; Ern et al., 2018) and references therein. Additionally, we apply a Savitzky-Golay polynomial smoothing on the spectral components in both meridional and vertical direction. We use a fourth order polynomial on 11 points in the vertical and a third order polynomial on 25 points in the meridional dimension. Inverse FFT from the spectrum yields a 3-D background field containing the large-scale signals. The temperature perturbations are then defined by subtracting this background point-wise from the original temperature field.





Zonal spectral filtering is an intuitive solution for a background removal in regularly sampled, global data sets such as general circulation model outputs of free-running models as well as assimilation products like reanalyses. The temperature and wind fields along the latitude circle are naturally periodic and with a regular (ideally high-resolution) sampling, an FFT analysis is directly applicable.

### 3.3  Time-Horizontal filtering for limb-sounder data

The background removal from Ern et al. (2011) is especially taylored to the sampling pattern of limb sounders on LEO satellites and was applied to SABER and HIRDLS (High Resolution Dynamics Limb Sounder) in the GRACILE climatology (Ern et al., 2018).

SABER provides a global coverage over the course of one day. Applying a similar background removal as for a global model data set hence, is a reasonable approach. However, data at a given latitude from the ascending or descending orbits,

respectively, are collected at the same local time rather than at the same universal time considering periods as short as a few days. This is the case because the orbit of the TIMED satellite is only slowly precessing. The period of one day, which SABER requires to collect the data for one global coverage, is close to the shortest period of planetary waves such as the quasi two-day waves in the mesosphere (e.g. Ern et al. (2013)) or fast modes in the southern polar vortex. Accordingly, the phase shifts in this observation period and the observed structures are not longer cyclical in longitude direction. In addition, there are gaps

between the measurement tracks. Satellite data hence require a more complex background removal including the temporal development.

The observation geometry of an LEO satellite limits the range of zonal wavenumbers and frequencies in which space-time spectra can be uniquely obtained (Salby, 1982). These limits allow to capture the periods of all Rossby wave modes (Salby, 1984) and equatorial wave modes (Kim et al., 2019). Furthermore, space-time spectra within the geometry limits have been

proven to be mathematically correct for all permitted wavenumbers and frequencies (Salby, 1984). The use of time-longitude spectra also prevents influences of day-to-day variations in the atmospheric background due to short-period travelling planetary waves (Ern et al., 2016, 2018). Due to the characteristic scales of inertial instabilities (long horizontal, short vertical and long temporal with respect to the gravity wave spectrum), they should be easier separable from gravity waves considering the zonal and time dimension than the vertical dimension.

Ern et al. (2011, 2013) have developed such a dedicated background removal for limb-sounding satellite temperatures taking both the zonal and temporal dimension in consideration for the spectral analysis. With the relatively low resolution of the satellite temperatures in longitude and time, gravity wave signatures appear as a uniform white-noise background level in longitude-time spectra. This white-noise spectrum has to be retained after the background removal. Global-scale waves, on the other hand, exceed this white noise level and should be removed. Ern et al. (2011) used a factor of 5 above the average

squared spectral amplitude of the white-noise spectral floor to define a threshold for separating between gravity wave signal and global-scale waves. The method uses overlapping 31-day time windows shifted by 15 days in each step. Spectra include zonal wavenumbers 0-6 and frequencies up to 0.7 cycles per day. In order to compute the spectra, the SABER profiles are interpolated in the vertical and along the observation tracks onto a fixed altitude-latitude grid.



In order to determine the residual temperature fluctuations, the background is reconstructed for the precise location and time
of each observation. First, spectra are interpolated in altitude and latitude. The background for the observation longitude and
time is then evaluated as the superposition of the single, global-scale waves with significant spectral amplitudes using their
spectral amplitudes and phases. In addition, the mean background temperature defined as the 31-day average plus a linear
trend is taken into account. Eventually, the method defines the temperature perturbation at a particular location and altitude
by subtracting the average local background temperature (averaged from the mostly two background temperature estimates)
from the overlapping time windows. In addition, in a second step the most relevant tidal modes in stratosphere and mesosphere
are explicitly removed (Ern et al., 2013). Lastly, an additional broad correction is applied that helps to calculate follow-
up parameters such as gravity wave momentum flux: a dominant vertical oscillation with vertical wavelength $\lambda_z \geq 40\mathrm{km}$
is removed to correct for remnants of quasi-stationary planetary waves and to limit the gravity wave spectrum to vertical
wavelength short enough to be covered by the momentum flux estimation procedure.

We apply this method as a horizontal background removal to SABER profiles in the rest of this study.

### 3.4 Diagnostic quantities

Linear gravity wave theory bases on the assumption that the total state of any atmospheric variables, e.g. temperature $T$, can
be defined as a sum of a background state, $\overline{T}$, and gravity wave perturbations, $T'$, like

$$T(\boldsymbol{x},t) = \overline{T}(\boldsymbol{x},t) + T'(\boldsymbol{x},t). \tag{6}$$

The background state is the superposition of a mean temperature profile and the large-scale influences like planetary wave
and synoptic inertial instability conditions at each instance in space $\boldsymbol{x} = (lon,lat,alt)$ and time $t$.

The variance $var(T')$ of the zonal mean residual temperature perturbation $T'$, also referred to as gravity wave variance, has
been used as a diagnostic quantity for gravity wave activity in measured and modeled atmospheres early on (e.g. Fetzer and
Gille (1994)) and is also the basis for our analysis.

$$var(T') := \frac{1}{N-1} \sum_i \left( T'_i - \langle T' \rangle \right)^2 \tag{7}$$

In Eq. 7, triangular brackets $\langle . \rangle$ represent a zonal mean at a specific snapshot or time range and $N$ stands for the number of
averaged values. This quantity shows "true" gravity wave structures in the data, if the background removal can be considered a
good estimate for the large-scale fluctuations.

In addition, the zonal mean temperature perturbation $\langle T' \rangle$ is defined as

$$\langle T' \rangle := \frac{1}{N} \sum_i T'_i. \tag{8}$$

In the previous subsections, we have introduced two different general methods to separate temperature perturbations from a
background that we want to apply in this study - horizontal and vertical spectral filtering. Resulting temperature perturbations
$T'$ are used to calculate more complex diagnostics like gravity wave potential energy densities and gravity wave momentum
fluxes to create climatologies of gravity wave activity.





270     In the case of horizontal filtering, in particular zonal wavenumber filtering, $\langle T' \rangle$ is expected to contribute only negligibly to the gravity wave variance, since the zonal average of the temperature state was already subtracted in the definition of the gravity wave temperature perturbations. After vertical filtering, $\langle T' \rangle$ is not necessarily small, since the perturbations were estimated in a way that is independent of the zonal structure of the background state. In this situation, $\langle T' \rangle >> 0$ gives a good measure for background state remnants in the derived perturbations, as shown by Rapp et al. (2018b).

275     In the ideal case, however, $\langle T' \rangle$ is zero or at least very small in comparison to the gravity wave perturbation that are analysed. Then, the zonal mean sqaured temerpature perturbation $\langle T'^2 \rangle$ is representative for the gravity wave variance.

$$\langle T'^2 \rangle := \frac{1}{N} \sum_i (T'_i)^2 \simeq var(T')|_{\langle T' \rangle = 0} \tag{9}$$

    The advantage of using $\langle T'^2 \rangle$ rather than $var(T')$ is that it is consistent with local averages such as used in global maps or spectra and which cannot rely on a zonal mean average.

## 4    Results and Discussion

### 4.1    A remnant signal threshold for a "successful" background removal

For evaluating gravity waves in real or realistic data, it would suffice to check, if the spurious inertial instability remnants in the estimated perturbations are smaller than the targeted gravity wave signals. However, the quantification of the gravity wave signal is the purpose of the analysis itself and, hence by definition, not known a priori. It therefore cannot be used as a reference 285   for successful background removal.

    For the removal of the background from real observations, it is necessary to define a success criterion. In our approach, the background with respect to gravity waves contains global scale variations caused e.g. by planetary waves and signals from inertial instabilities. We consider a background removal as successful, if the background remnants are smaller than the errors introduced by the measurement method and the retrieval. The error budget of an instrument is usually described by precision 290   and accuracy, where accuracy describes "systematic" errors which are present in a similar magnitude on the entire data set and precision refers to the noise-like errors that affect a measured parameter randomly. In the observation of fluctuations, like gravity waves, the uncertainty is closer linked to the precision than to the accuracy of the instrument, since errors characterized by accuracy are removed together with the background. The magnitude of both accuracy and precision usually vary with altitude. Based on this approach, we use the minimal precision value of the SABER instrument reported for the stratosphere of 295   0.3K (Ern et al., 2018, and references therein) as a threshold for a successful background removal. For most of the stratosphere and for the mesosphere, this threshold is more conservative than the targeted detection limit for gravity waves.





## 4.2 Removing artificial inertial instability signals: Minimum appropriate filter cutoff for vertical and horizontal background removals

A spectral background removal will usually remove global-scale variations gradually, i.e. each global-scale process is described
300  by several spectral components, and the more wavenumbers are removed the more of the background signal is removed as
well. The success criterion defined above then allows us to identify threshold characteristic vertical wavelengths or zonal
wavenumbers for horizontal or vertical background removals, respectively. In particular, this can be directly applied on the
synthetic data. On our artificial inertial instability data set we can test the different approaches and find the criterion for a
successful background removal matched when the mean squared temperature perturbation (cf. Eq. 9) falls below $0.09 \, \mathrm{K}^2$.

305    Figures 2 and 3 shows altitude-latitude cross sections of the zonal temperature variance (cf. Eq. 7) for the artificial inertial
instability data (see Sect. 2). In Fig. 2, vertical filtering is applied with decreasing cutoff vertical wavelengths and, in Fig. 3,
horizontal filtering with increasing cutoff zonal wavenumbers, respectively. The layered "pancake" structures are a persistent
feature, but eventually both methods, vertical filtering at cutoff wavelength $6 \, \mathrm{km}$ (see Fig. 2(f)) and horizontal filtering at cutoff
wavenumber 7 (see Fig. 3(f)) remove the signal with remaining remnants of less than $0.05 \, \mathrm{K}^2$ magnitude. This value is even
310  smaller than the threshold of the success criterion we defined in Sect. 4.1.

    We chose most of the cutoff limits to match values used in previous satellite studies. Fig. 2 (a) shows the structure after
removing vertical wavelengths down to $15 \, \mathrm{km}$, as used by Rapp et al. (2018b). A similar level of removal is reached by
applying cutoff zonal wavenumber 0 in the horizontal filtering case (see Fig.3 (a)). A cutoff wavelength of $10 \, \mathrm{km}$ as used
in Fig. 2(b) was used before by Tsuda et al. (2000) and Alexander and Barnet (2007) for vertical filtering. However, it has
315  been shown that this cutoff removes a significant part of the gravity wave spectrum important for the middle atmosphere
(Preusse, 2001; Preusse et al., 2008; Alexander et al., 2010). In our case, horizontal filtering with cutoff zonal wavenumber
1 (see Fig. 3 (b)) gives a similar temperature variance result. On the other hand, Alexander et al. (2008a) applied horizontal
filtering with a cutoff zonal wavenumber 3 using an S-transform approach. Fig. 3 (c) shows the temperature variance structure
after removing zonal wavenumber 3. On the vertical filtering side, a similar result is achieved with a cutoff wavelength of
$9 \, \mathrm{km}$ (see Fig. 2(c)). The values of cutoff vertical wavelength $8 \, \mathrm{km}$ or cutoff zonal wavenumber 4 were not used in previous
studies, but the results show an intermediate value for the remnants (see Fig. 2(d) and 2(d)). The horizontal filtering with
cutoff wavenumber 6 (see Fig.3 (e)) is the first to fall below the precision threshold of $0.09 \, \mathrm{K}^2$. A number of studies used this
cutoff previously (e.g. Fetzer and Gille, 1994; Preusse et al., 2002; Ern et al., 2018), since zonal wavenumber 6 is close to
the sampling limit for infrared sounders on LEO. The vertical filtering variance magnitude counterpart here is found at cutoff
wavelength of $7 \, \mathrm{km}$ (see Fig.2 (e)). Finally, with vertical cutoff at $6 \, \mathrm{km}$ and horizontal cutoff at 7, the zonal variance decreases
even to $<0.05 \, \mathrm{K}^2$ (see Fig.2 (f) and 3 (f)).



### 4.3 Compromise between removing inertial instability remnants and preserving gravity wave signals in ERA5 temperatures

The upper row of Fig. 4 shows zonal mean structures of temperature perturbations (cf. Eq. 8) from ERA5 data on 3 December
2015, 00:00 UTC, after applying different filterings. As an indication for significance, we also show the standard deviation of
the zonal means, calculated as $\sqrt{var(T')}$ (cf. Eq. 7), in the lower row. The zonal mean temperature perturbation for vertical
filtering with cutoff wavelength 15 km (cf. Fig. 4(a)) displays strong layered bands of vertically alternating positive and
negative peaks at the equator, i.e. spanning from 20 °S to 20 °N, as well as in the midlatitudes approximately between 30
and 45 °N. This shows the presence of the inertial instability also in the ERA5 data. Equatorial values are 180 °-phase-shifted
with respect values at the midlatitudes, as expected from theory (Dunkerton, 1981) and found in previous inertial instability
observations (Hayashi et al., 1998; Smith and Riese, 1999; Rapp et al., 2018b). The midlatitude stack widens in meridional
extent with altitude between 20 and 45 km, while the tropical feature narrows. The main pancake structures are significant, i.e.
larger than the corresponding standard deviations.

The other two panels in Fig. 4 show the zonal mean temperature perturbations for the cutoffs, which provide the best removal
of our artificial inertial instability signals (see Sect. 4.2). In the middle (Fig. 4(b) and (e)), we applied vertical filtering with
a cutoff wavelength of 6 km. The mean perturbations still show layered structures, in particular in the latitude bands where
the pancake structures were found for a cutoff at 15 km. The magnitudes, however, are less than 1 K and overall insignificant.
In the tropics (approximately from 20 °S to 20 °N), higher values in the standard deviation point to signals that have smaller
vertical scales than the removed inertial instability. The symmetric structure around the equator with a smooth, Gaussian-like
peak and a half-width of approx. 10 ° indicates that this maximum may be attributed to Kelvin waves (Smith et al., 2002). On
the right (Fig. 4(b) and (e)), we applied horizontal filtering with a cutoff at wavenumber 6. Except for a few spots above the
tropical tropopause (around 5 °S and below 25 km) and at high latitudes (north of 60 °N) and high altitudes (above 35 km),
the zonal mean does not show structures exceeding 0.01 K. The standard deviation, however, shows enhancement at higher
altitudes and in particular in a structure around the stratospheric polar vortex, which indicates real gravity wave activity. This
feature is also present in the two other standard deviation plots and will be discussed in more detail later on.

We have seen already that the quality of a spectral filtering is highly dependent on the cutoff length scale used. Ideally, a
transition between mesoscale and synoptic-scale fluctuations (previously referred to as "spectral gap") can be identified, where
the temperature variance will be insensitive to the cutoff length separating gravity wave activity at small- and meso-scales from
global-scale inertial instability signals. However, such a transition is not necessarily present in the stratospheric temperature
variance, and very unlikely to be found in the troposphere.

With its dense regular sampling, the realistic global ERA5 temperature data allow a sensitivity study with respect to temper-
ature variances on the cutoff length scales for a wide range of cutoffs of both vertical and horizontal filtering. Figure 5 shows
the dependence of temperature variances on the background removal. The variances are calculated using Eq. 9, but here the
average is defined over a domain from 90 °W to 45 °E longitude, 30 to 45 °N latitude and 20 to 50 km altitude instead of
one latitude circle. This region is chosen, as Fig. 1 shows this domain to contain the largest inertial instability signal. The two





panels show the scaling behaviour for (a) vertical filtering with cutoff vertical wavelength in the range 5 to 20 km and (b) horizontal filtering from cutoff zonal wavenumber 0 to 42. The yellow curve shows results for ERA5 data at 3 December 2015 at 00:00 UTC and the blue curve for the artificial inertial instability constructed as described in Sect. 2.3. We use the blue curve as a reference of how a pure inertial instability signal is affected by the background removals.

Different sensitivities of the mean temperature variance for more rigorous filtering are evident. For vertical cutoff wavelengths larger than 15km, the averaged variance of ERA5 temperatures increases slowly with a linear gradient. In the same cutoff wavelength range, the variance stays mostly constant in the artificial inertial instability perturbations. The gradient becomes gradually steeper in both data sets between 15 and 10 km. In this range, where both data sets show similar behaviour, the averaged temperature variance is likely dominated by the inertial instability signal. The gradient for the ERA5 data remains

similar also for cutoff vertical wavelength shorter than 10 km while for the artificial inertial instability data the decrease continues to steepen. This different behaviour indicates a switch to gravity waves as the dominant dynamics process, missing in the pure inertial instability perturbation data.

The sensitivity of temperature variance to increasing cutoff zonal wavenumbers shows a range of cutoffs where both the realistic ERA5 data as well as the artificial inertial instability perturbations behave in a comparable way. From cutoff zonal

wavenumber 1 to 6, both curves of averaged temperature variances are decreasing fast, indicating the dominance of the inertial instability signal. At wavenumber 6 there is an abrupt change in the variance decrease in ERA5 data, and the slope continues to change between wavenumber 6 and 12. At wavenumbers above cutoff zonal wavenumber 12 the gradient is about constant. In comparison, the variances decrease of the artificial perturbations continues up to wavenumber 12 and then directly switches to stagnation at extremely small averaged temperature variances. That the averaged temperature variance for the synthetic inertial

instability stays constant at wavenumbers 2 and 3, wavenumbers 5 and 6 and wavenumbers 10 and 12 in the artificial data is likely connected to the regular structure of the modeled inertial instability, which spans over approximately a third of the latitude circle.

Similar to the vertical filtering case, we interpret similarities and differences in the variances of the two data sets. The variance is likely dominated by the inertial instability influence up to wavenumber 6. For wavenumbers larger than 7 gravity

waves are taking over as the most dominant process, though up to wavenumber 12 there may be remnants of the inertial instabilty. The slow decrease at high wavenumbers shows that eventually energy is taken out of the gravity wave scales as well.

### 4.4 Situation in a climatology

The diagnostics shown for gravity wave climatologies from measurement or model data are usually energy quantities like gravity wave potential energy densities (Tsuda et al., 2000; de la Torre et al., 2006; Rapp et al., 2018b) or gravity wave

momentum fluxes (Ern et al., 2004; Wang and Alexander, 2010; Alexander, 2015; Ern et al., 2018). These quantities are not calculated from gravity wave variances (Eq. 7), but using a spatial average of the the squared temperature perturbation (Eq. 9). Figures 6 and 7 show the zonal mean structure of ERA5 squared temperature perturbations for vertical and horizontal filtering employed with different scales of the background removal. Similar to Figs. 2 and 3, the different cutoff length scales are mostly chosen according to levels that have been used in previous studies. For the vertical filtering, we limit the presented results to





cutoff vertical wavelengths of 15 km (Fig. 6(a)), 10 km (Fig. 6(b)) and 6 km (Fig. 6(c)). The first two were used in studies by other groups before, while the 6 km cutoff marks the transition point in our sensitivity curve (Fig. 5) where the gravity wave signal dominated the temperature variance in the stratosphere. For the horizontal filtering, we used cutoff zonal wavenumbers at 0, 1, 3, 4, 6, and 7 as used before for the artificial data. Furthermore, we add the higher order cutoffs at wavenumber 12, 18 and 42 to the figure to evaluate how the structures are changing if more wave components are removed then necessary to

eliminate the inertial instability structures.

    In the following, we concentrate on four structural features and their evolution under different cutoffs in the two background removals. First, Fig. 6(a) shows a typical pancake structure in the midlatitude stratosphere (approximately between 30 and 45 °N and above 25 km altitude) associated with the inertial instability (Rapp et al., 2018b). A corresponding but isolated stack is present above the equator between 30 and 40 km altitude.

The isolated structures merge into one broad sheet of perturbation at altitudes a few kilometers below the stratopause. Decreasing the cutoff wavelength in the vertical filtering reduces the pancake structure at midlatitudes in a way that it is not further recognizable in the results from 6 km vertical filtering. The tropical band of the structure is more persistent while being reduced in magnitude, but remaining especially strong at 35 to 40 km altitude. In addition, a more rigorous vertical filtering seems to narrow the vertical distance between the variance peaks. As discussed in Sect. 4.3, the remaining structures may be

caused by tropical wave modes and, in particular, by Kelvin waves.

    In comparison, the horizontal filtering results show less evidence for the typical pancake structure, both at midlatitudes and in the tropics. In the tropics, above 30 km altitude some layered structures emerge for low cutoff wavenumbers (cf. Fig. 7(a) and (b)). The midlatitudes, however, are dominated by a strong ($> 10 K^2$), large-scale perturbation structure that does not show the stacked pancake features and is likely connected to remaining Rossby wave signals. At cutoff zonal wavenumbers 3 and

4 (cf. Fig. 7(c) and (d) respectively), the Rossby wave feature is mostly removed and some pancake-like structures are found in the midlatitudes which are already reduced compared to the vertical filtering with cutoff wavelength 15 km. In the tropics, no clear pancake structure remains increasing the cutoff zonal wavenumber further, but a small local maximum in Fig. 7 (e) at 40 °N and 30 km altitude could still be associated with the pancake structures.

    At altitudes above 20km, the general structure continues to decrease in magnitude, but remains very similar in structure for

wavenumbers 6 and larger. This is consistent with gravity waves at a wide range of different scales causing these features. Particularly pronounced is a second structure between 60 °N and 80 °N and 30 to 50 km altitude, which shows still enhanced perturbations at wavenumber 42 (cf. Fig. 7 (i)). This is likely associated with gravity waves in the winter polar vortex, and Rapp et al. (2018b) attribute a major part of the feature to gravity wave sources at Greenland. The structure is also found in the vertical filtering results, but it extends less far into higher altitudes as well as less wide equatorwards. In general, the

perturbation structures after vertical filtering show a few degree latitude wide gap of comparably small perturbations around 60 °N. This anomaly is located approximately in the center of the polar night jet where we expect a predominance of long vertical wavelength of gravity waves. The feature is not present in the horizontal filtering results. In fact, at this location, horizontal filtering indicates a maximum of squared temperature perturbation.





For very low cutoff zonal wavenumbers, the high-latitude maximum is concealed by a larger perturbation due to Rossby
waves in middle and high latitudes. From cutoff wavenumber 6, it is not distorted by pancake-like, overlaid structures anymore.
Then, the overall shape of the enhanced temperature perturbation distribution is not changed up to a cutoff wavenumber 18
and also keeps roughly the same magnitude. This indicates that the horizontal wavelengths of the waves amount to at most
∼1000 km as Coriolis force arguments for gravity waves at these latitudes would suggest (Alexander et al., 2002; Preusse
et al., 2006). The very strong horizontal filtering with cutoff zonal wavenumber 42 still shows the feature but with decreased
magnitude, which indicates that part of the gravity wave spectrum is also removed by the background removal.

Below 20km, the tropical tropopause and the tropopause inversion layer show up distinctively in the vertically filtered data,
depicted by a enlongated, strong perturbation signal spanning all the way from 30 °S to 30 °N between 15 and 25 km (cf. e.g.
Fig. 6(a)). Rapp et al. (2018a) explain this third structure with shortcomings of the vertical filtering to identify background
temperature structures like the tropical tropopause with the short vertical length scale. Even a rigorous vertical filtering with a
cutoff wavelength of 6 km retains the signal. This problem is similar to the misinterpretation of inertial instability perturbations
due to their small vertical wavelength. In comparison, the horizontal filter is removing a large part of the tropopause signal
from the perturbations already with cutoff wavenumber 0, i.e. only removing the zonal mean. This could be explained by the
tropical tropopause and the tropoause inversion layer being rather stable in altitude over all longitudes. However, even at zonal
wavenumber 42, the remnant structure is still larger than the gravity wave fluctuations above and below, indicating that the
remnants of the tropopause still masks the gravity wave signal.

Finally, all of the results show enhanced temperature perturbations in the troposphere below 15km around 40 °N. With
cutting more rigorously, both filtering methods reduce the zonal mean magnitude of this feature, but it is never fully removed,
even with high horizontal wavenumber cutoffs of 18 or 42. The dynamically active region of the troposphere is very difficult for
all gravity wave background removals due to the large and possibly abrupt variation of the temperature over small distances. In
the troposphere a wide spectrum of Rossby waves up to high zonal wavenumbers exists, which is filtered around the tropopause
according to the Charney-Drazin criterion (Charney and Drazin, 1961). Accordingly, the stratospheric background atmosphere
is dominated by planetary waves with zonal wavenumbers smaller than 4 (e.g. Domeisen et al., 2018; Barnett and Labitzke,
1990; Pawson and Kubitz, 1996), consistent with the findings in this paper. In order to investigate altitudes below 20 km
from LEO infrared sounders, higher wavenumbers than wavenumber 6 have to be taken into account. Even at high zonal
wavenumbers, the results will need further validation, which, however, is beyond the scope of this study.

## 4.5 Case and one-year time series of SABER

So far we have considered a case study for a particular, strong inertial instability event. In order to evaluate whether the results
are representative, we investigate one year of SABER data.

As already addressed in Sect. 2 and 3, SABER data is much sparser than ERA5 and we have to rely on a more complex
horizontal filtering approach. The method is taylored to the regular global sampling pattern of orbiting limb-sounders with
measurement tracks quasi-parallel to the satellite orbit track. Satellite observations with different sampling methods may need
slightly different approaches. For instance, longitude-time spectra can be calculated from GPS-RO profiles; this has been done



by Alexander et al. (2008b), for the analysis of Kelvin waves using COSMIC data. Such spectra could be used for background removal, but more care would have to be taken because the number of data points available for a given latitude will vary

strongly.

Like Fig. 4 for ERA5, Fig. 8 shows the zonal mean temperature perturbations and corresponding standard deviations now from SABER data. On the left (Fig. 8(a) and (c)), we applied vertical filtering with cutoff wavelength of 15 km. Again, we find the pancake structure in the tropics and midlatitudes charcteristic of the inertial instability. Since SABER looked southward on 3 December 2015, we do not have data coverage at high latitudes for this case. The standard deviation show layered structures

as well, which are especially strong in the midlatitudes. On the right (Fig. 8 (b) and (d)), we show the results from the time-horizontal filtering for the SABER data using cutoff zonal wavenumber 6. Over the course of one day, SABER data only contains 32 measurement points along one latitude circle and thus provides much weaker statistics than the ERA5 data. This presumably explains remnant values of mean temperature perturbations in the SABER data, which, however, do not show the characteristic "pancake" structure typical for inertial instabilities and are mostly below 0.5 K. In the standard deviation, we see

more activity in the winter hemisphere than in the summer hemisphere. This is consistent with the general understanding of source activation and global distributions of gravity waves (Fritts and Alexander, 2003; Geller et al., 2013). Overall, magnitudes are lower than in Fig.8 (a) and (c). In vertical filtering, most of this distribution was distorted by the large, layered features of the inertial instability.

Figure 9 shows daily averages of mean zonal squared temperature perturbations (cf. Eq. 9) in a midlatitude box from 90°W

to 45°E and from 30°N to 45°N for one year of SABER data (July 2015 to June 2016). The upper panel shows results from a vertical Butterworth filter with 15 km cutoff wavelength, and the lower panel presents results form horizontal background removal with a cutoff zonal wavenumber 6.

Throughout the entire year, the vertically filtered data shows a pronounced signal above the tropopause, i.e. between 20 and 25 km, and around the stratopause, i.e. between 40 and 50 km altitude. Furthermore, this data set exhibits remnant pancake

structures persisting for several days through the northern hemisphere winter (December to mid February). This is consistent with favourable conditions for inertial instability particularly around the winter solstice (Rapp et al., 2018b).

After horizontal filtering some wave-like structures are remaining, however smaller in magnitude and of shorter duration. These structures indicate very active gravity wave events dominating the average in the domain. Also, the horizontal background removal does not show the very strong signals associated with the tropopause and stratopause.

## 5   Summary and conclusion

The analysis of satellite data for gravity waves relies on a scale separation between the targeted gravity waves and larger scale structures. Recently, concern was raised that inertial instabilities might not be properly removed in generating gravity wave climatologies and thus cause spurious signals in the derived distributions (Rapp et al., 2018b; Harvey and Knox, 2019). Therefore, the subject of this study is to assess the capability of horizontal and vertical spectral background removals in respect





to isolate gravity waves from inertial instability effects. In addition, we aim at determining optimal cutoff length scales for the background removal.

We approached this by considering a particularly strong inertial instability event on 3 December 2015, which was discussed in the study of Rapp et al. (2018b). As test data, we use ERA5 reanalysis temperatures, SABER temperature observations and a synthetic data set containing only an idealized inertial instability signal. For the assessment we also need a criterion for

a successful background removal. The background removal is considered successful if the gravity wave variances dominate over the remnants of the removed inertial instability. For practical purposes we also introduced as a second criterion that the background removal is considered successful if the remnants of the removed inertial instability are smaller than the precision of the instrument from which we determine the climatology. The advantage of this criterion is that the gravity wave distribution does not need to be known a priori.

First, we compared the vertical and horizontal filtering results for the idealized inertial instability signal. Zonal mean temperature perturbations and zonal mean variances show that both methods are in principle capable to remove an inertial-instability-like perturbation to a limit that it falls to a level below the SABER precision threshold. However, in vertical filtering we had to eliminate all fluctuation larger than 7 km in the vertical to achieve a sufficient signal reduction. This would lead also to the removal of a major part of the gravity wave spectrum. A comparable level of removal after horizontal filtering was achieved

with a cutoff zonal wavenumber of 6, which is no restriction to the gravity wave analysis, in particular if momentum flux is considered.

The investigation of ERA5 data shows that for the midlatitude stratosphere gravity waves dominate over inertial instability signals for vertical cutoff wavelengths shorter than about 10 km and for horizontal cutoff wavenumbers larger than 6. After vertical background removal remnants of the stratopause remain. In addition, the tropical stratosphere showed perturbations

over all altitudes from 20 to 45 km that indicate Kelvin wave activity. Horizontal filtering, in contrast, removes both effects. Horizontal filtering is also more capable in dealing with the tropical tropopause, but remnants larger than the average gravity wave variance remain. In the subtropical troposphere and lowermost stratosphere zonal wavenumbers larger than 18 are required to reduce the signal of Rossby waves. Further consideration will be needed in future, if data sets should be evaluated for such lower altitudes, as well. A zonal wavenumber filter applying wavenumbers higher than 18 removes already part of the

gravity wave spectrum.

A one year analysis of SABER data reveals inertial instability signals for the vertical filtering, but no evidence for inertial instabilities after horizontal filtering. Hence, the case study of one inertial instability event is representative.

Considering existing satellite climatologies, such datsets which are based on time-longitude spectra including zonal wavenumber 6 are likely free of the influence of large scale inertial instabilities. This includes for instance the GRACILE climatology

(Ern et al., 2018) and studies based on this or similar data sets. Some influence is expected on data sets generated by vertical filtering with a 10 km cutoff wavelength, such as GPS investigations by Tsuda et al. (2000); de la Torre et al. (2006). In addition, an influence of Kelvin waves is expected for low latitudes. The relatively short cutoff wavelength applied in these studies, however, is also cutting deep into the gravity wave spectrum, in particular for high latitude winter conditions. Perturbations





based on vertical filtering with a 15 km cutoff wavelength on the other hand are dominated by inertial instability depending on
season and geographical region, as was already shown by Rapp et al. (2018b).

       In conclusion, for the removal of a background containing inertial instabilities and gravity waves, horizontal spectral filtering
with a zonal wavenumber of 6 or higher provides the best results. It is therefore recommended to use such an approach where
the data allow for it. For altitudes around the tropopause, larger zonal wavenumbers are required, but zonal wavenumbers larger
than 18 reduce the fraction of retrievable gravity waves in the stratosphere greatly.

*Data availability.* The ERA-5 data used for this study are freely available from (C3S). SABER data are freely available from GATS Inc. at
http://saber.gats-inc.com. Precision estimates for SABER temperatures are given on the SABER website at http://saber.gats-inc.com/temp_
errors.php and in Remsberg et al. (2008).

     *Competing interests.* The authors declare that they have no conflict of interest.

     *Acknowledgements.* The work of C. Strube was funded by the project Source Variability (PR 919/4-2) as part of the research unit MS-
GWaves (FOR 1898) of the Deutsche Forschungsgemeinschaft (DFG). In the same framework ME's work was partly funded by DFG (ER
474/4-2).



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

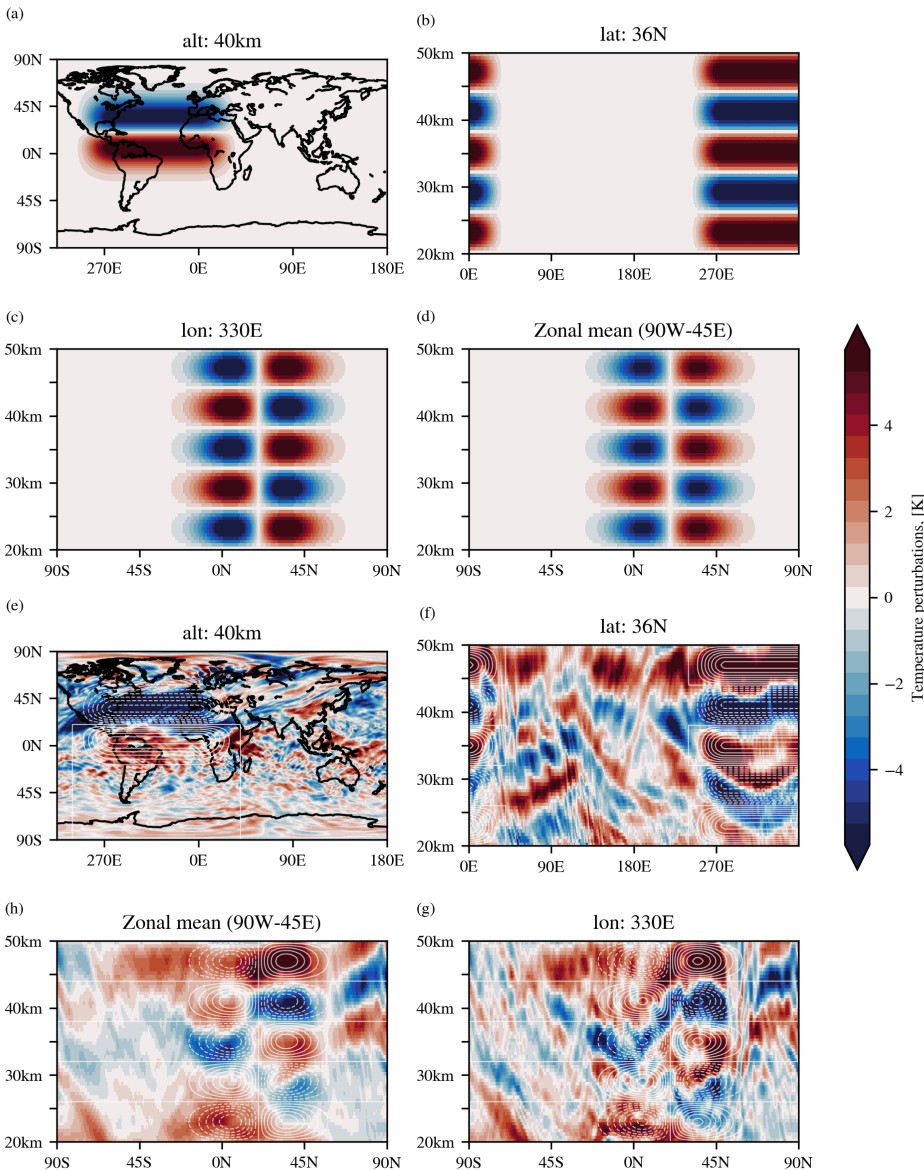

**Figure 1.** Cross sections of temperature perturbations of the artificial inertial instability, we defined in Sect. 2.3, are shown in the upper row: (a) A longitude-latitude cross section at 40 km altitude (black contours mark the coastlines), (b) a longitude-altitude cross section along the 36 °N latitude circle, (c) a latitude-altitude cross section along the 330 °E meridian and (d) the zonal mean in the longitude band from 90°W to 45°E (in the style of Fig. 1b) in Rapp et al. (2018b)). The lower row shows the respective plots as the upper row from the ERA5 temperature perturbation applying the fifth order Butterworth vertical filter with cutoff wavelength of 15 km. For easy comparison, the white contours in the lower row plots show the perturbation levels of the artificial data set in 2K steps (solid lines indicate positiv values, dotted lines negative values).





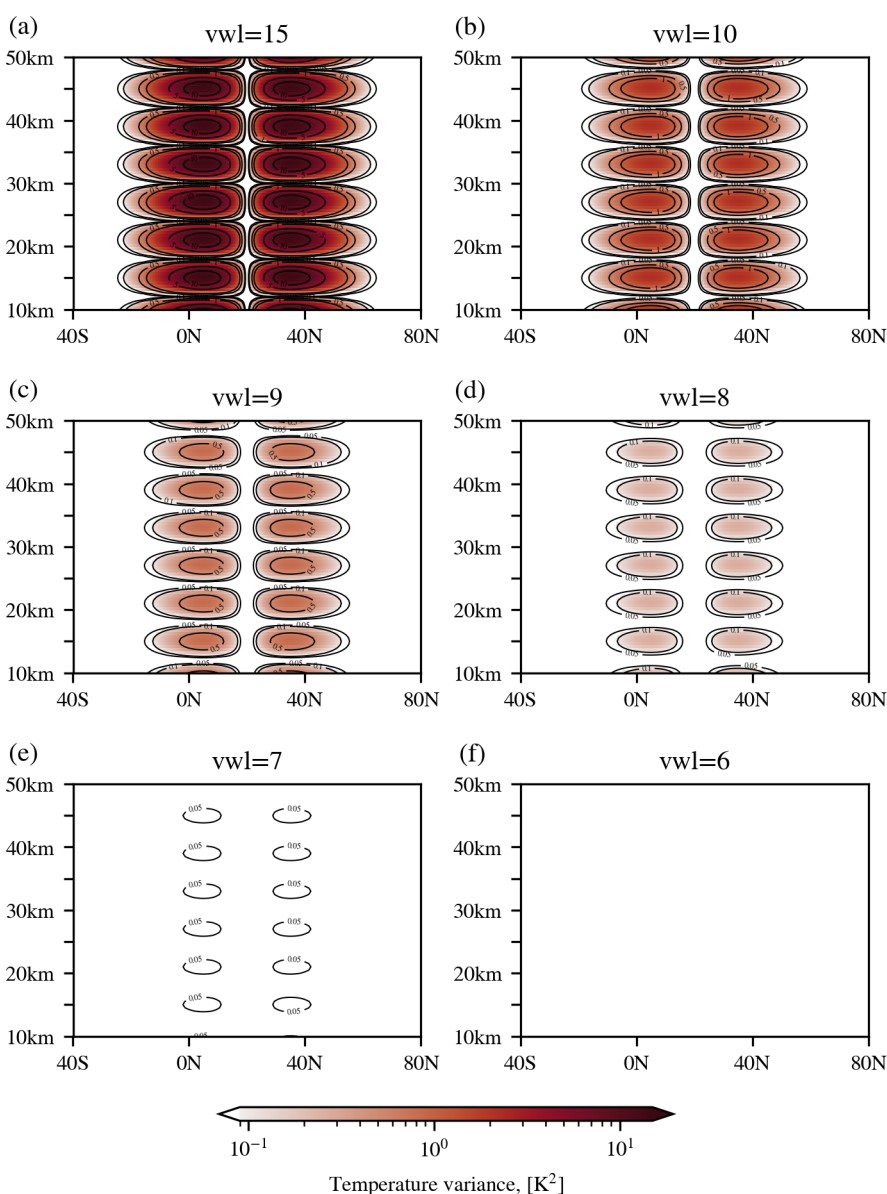

**Figure 2.** Zonal temperature variance of artificial inertial instability temperature perturbations after vertical filtering (fifth order Butterworth) with a cutoff wavelength of (a) 15 km, (b) 10 km, (c) 9 km, (d) 8 km, (e) 7 km and (f) 6 km.



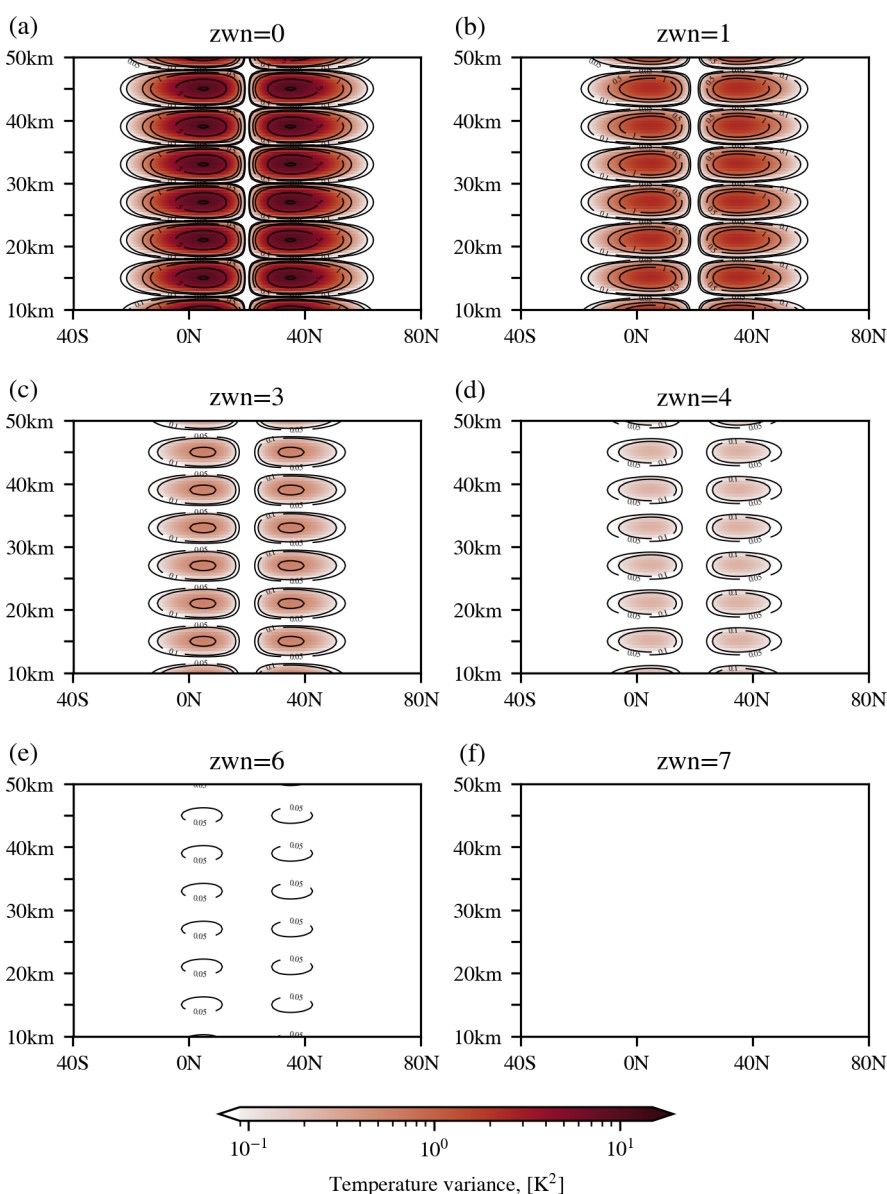

**Figure 3.** Zonal temperature variance of artificial inertial instability temperature perturbations after horizontal filtering (zonal mean with additional smoothing in meridional and vertical components) with cutoff zonal wavenumber (a) 0, (b) 1, (c) 3, (d) 4, (e) 6 and (f) 7.



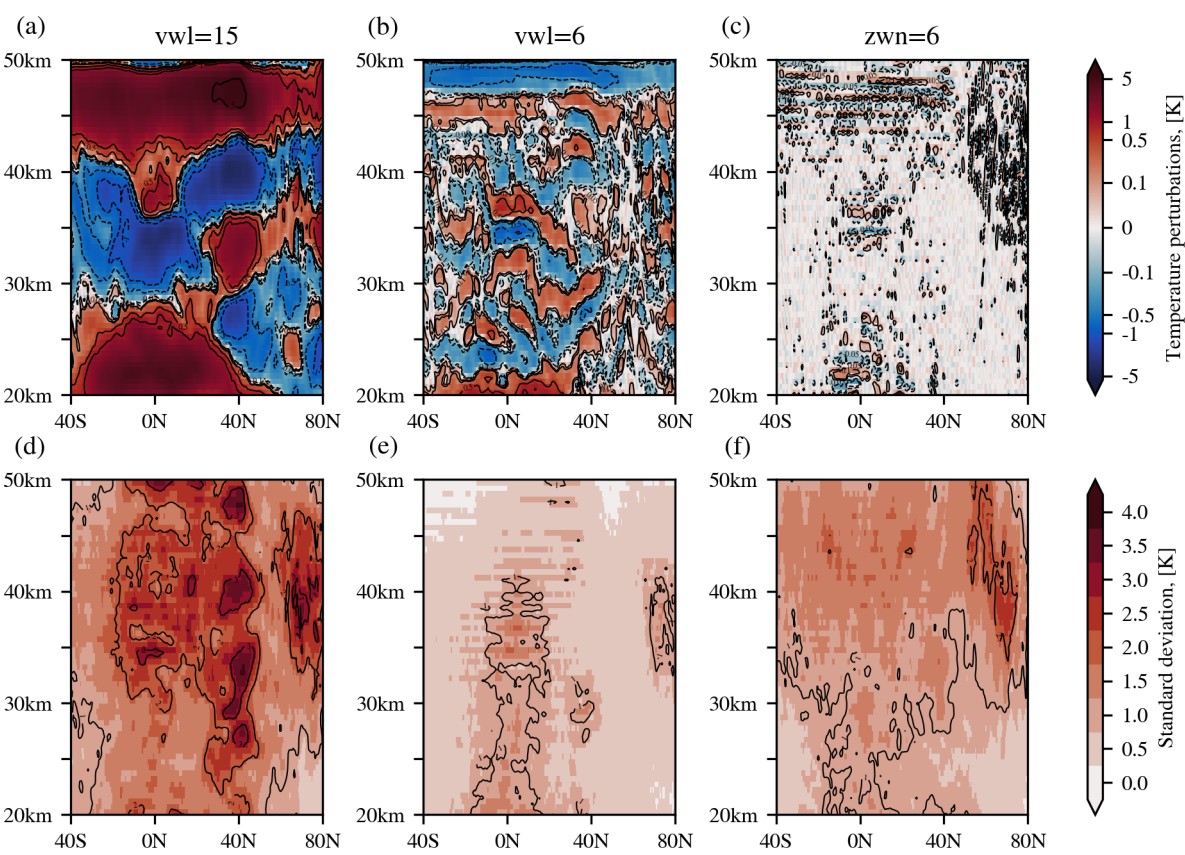

**Figure 4.** Zonal mean temperature perturbations from ERA5 temperatures for 3 December 2015, 00:00 UTC after (a) vertical Butterworth filtering with 15 km cutoff wavelength, (b) vertical Butterworth filtering with 6 km cutoff wavelength and (c) horizontal FFT filtering with cutoff zonal wavenumber 6 and additional vertical and meridional smoothing in the upper row. In the lower row (d), (e) and (f), we show the respective standard deviations of (a), (b) and (c). Contours show indicated temperature perturbation levels to highlight the structure.

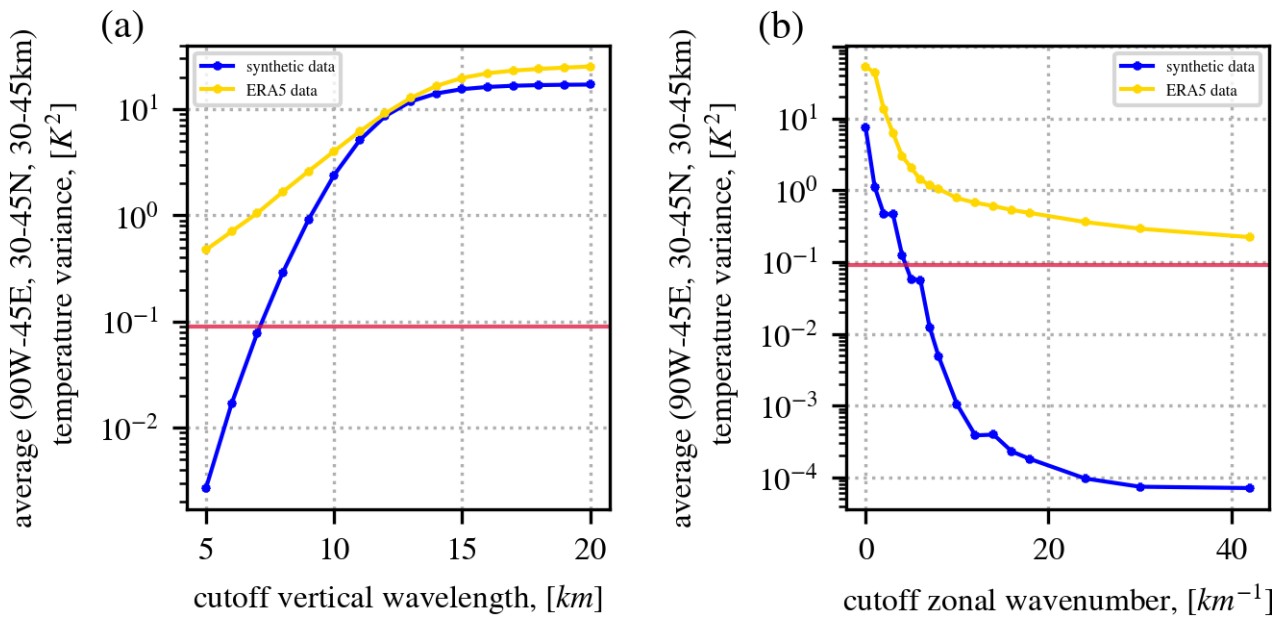

**Figure 5.** Dependence of mean temperature variance in a box of 90°W to 45°E longitude, 30°N to 45°N latitude and 20 to 50km altitude on the characteristic cutoff length scale for different filtering methods: (a) vertical fifth order Butterworth filter with cutoff in the vertical wavelength and (b) horizontal FFT filter with additional smoothing in vertical and meridional direction and cutoff in zonal wavenumber. In each plot, the blue line represents artificial inertial instability perturbations and the yellow line shows the corresponding results for ERA5 temperatures. The red horizontal line marks $0.09K^2$ as precision threshold of the SABER instrument.



# Vertical filtering

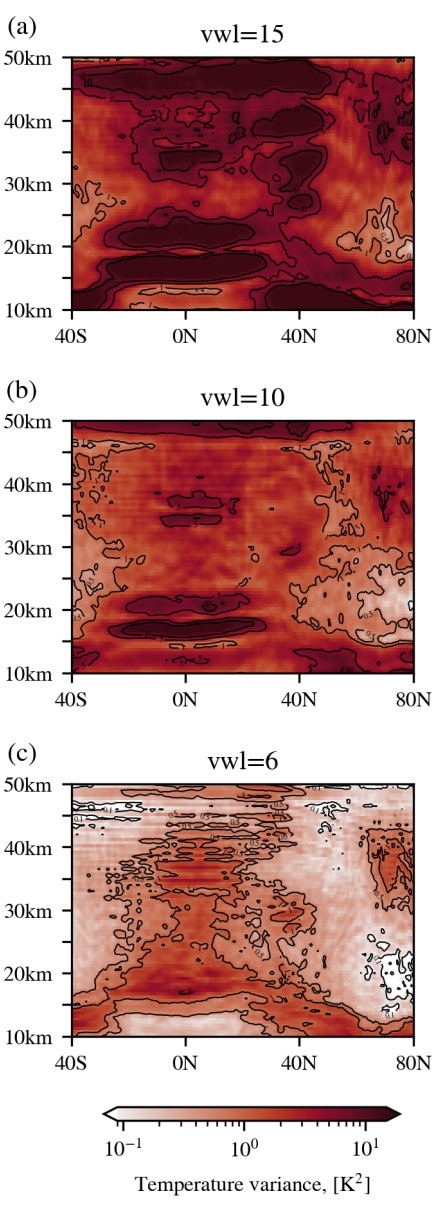

**Figure 6.** Vertically filtered (fifth order Butterworth) zonal mean squared temperature perturbations (cf. Eq. 9) with a cutoff wavelength of (a) 15 km, (b) 10 km and (c) 6 km of ERA5 temperatures on 3 December 2015, 00:00 UTC.



# Horizontal filtering

**Figure 7.** Horizontally filtered (with additional smoothing in meridional and vertical components) zonal temperature variance of ERA5 temperatures on 3 December 2015, 00:00 UTC, with cutoff zonal wavenumber (a) 0, (b) 1, (c) 3, (d) 4, (e) 6, (f) 7, (g) 12, (h) 18 and (i) 42.

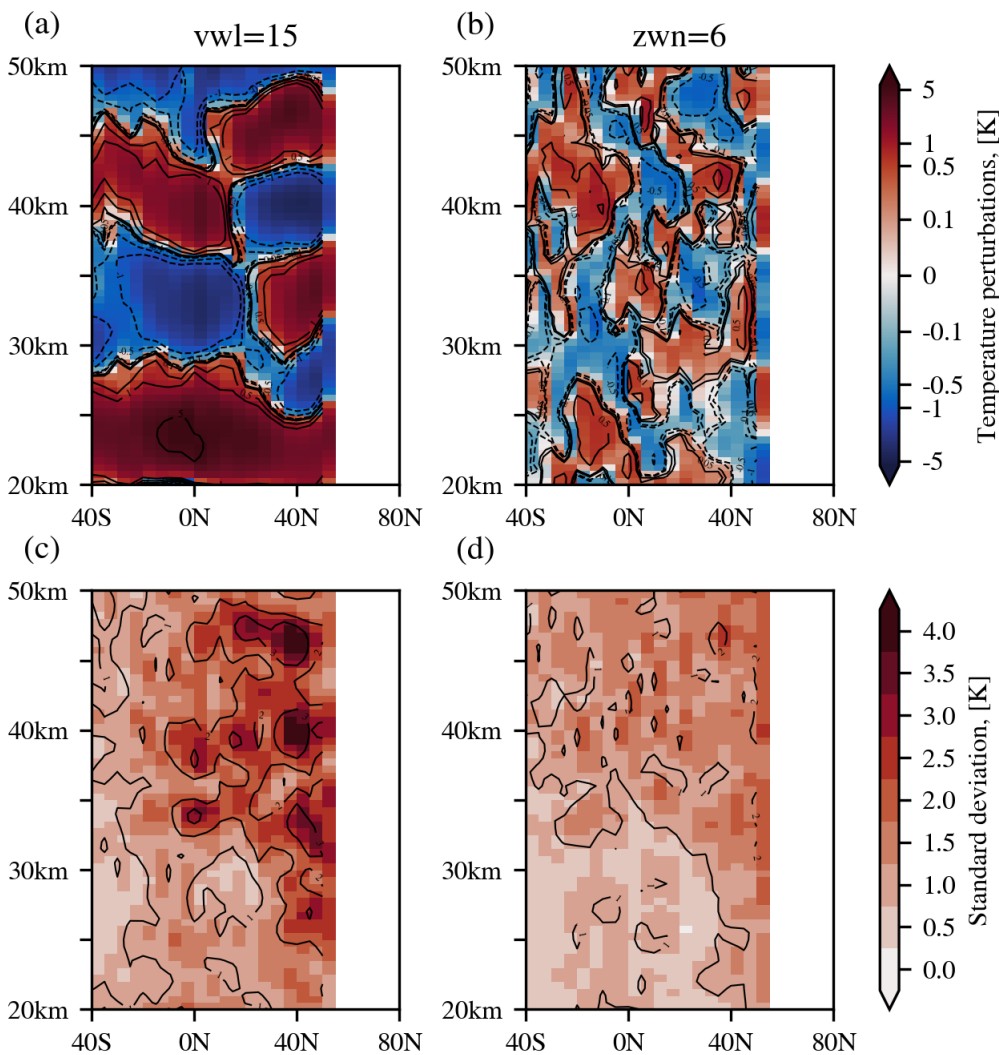

**Figure 8.** Zonal mean temperature perturbations and corresponding standard deviations of SABER data. The upper row shows the mean perturbation from (a) vertical filtering (fifth order Butterworth) with cutoff vertical wavelength of 15km and (b) horizontal filtering with the method from Ern et al. (2011) with cutoff zonal wavenumber 6 and period ≈1.4 d. The lower row shows in (c) and (d) the corresponding standard distributions of (a) and (b).





**Figure 9.** Time series of daily mean temperature perturbation profiles from SABER in the midlatitude box, 90°W to 45°E longitude, 30 to 45°N latitude, for one year (1 July 2015 to 30 June 2016) after (a) vertical Butterworth filtering with 15 km cutoff wavelength and (b) horizontal filtering (zonal mean with additional smoothing in meridional and vertical components) with cutoff wavenumber 6. Contours show the indicated value of daily mean box mean temperature perturbation.