# Peer review of "Removing spurious inertial instability signals from gravity wave temperature perturbations using spectral filtering methods"

_Atmospheric Measurement Techniques, 2020_

## Short Comment (SC1) · 30 Apr 2020

Dear authors,

This is not a review. I just noticed that on P3L63 you reference Šácha et al. (2016), but it should be either Šácha et al. (2015) or Šácha et al. (2014), which are observational studies.

Furthermore, I am really interested in your study and would like to ask some questions: 1) When using some simple form of vertical background separation (e.g. polynomial fit) is it possible to identify the potential presence of the inertial instability signal in the data

from analysis of the vertical wavenumber power spectrum density of disturbances? I.e. as long as the spectrum slope follows the theoretical saturated GW spectrum, then we are on the safe side?

2) As you acknowledge in the paper, all GW types are highly sensitive to the background winds during their lifecycle, can you exclude the possibility that a portion of GW activity can be filtered out using the horizontal filtering? How do you expect GWs to interact with the inertial instability region?

Thank you very much for your interesting paper and looking forward to your reply,

Petr Šácha.

Šácha, P., Foelsche, U., and Pišoft, P.: Analysis of internal gravity waves with GPS RO density profiles, Atmos. Meas. Tech., 7, 4123–4132, https://doi.org/10.5194/amt-7-4123-2014, 2014.

Šácha, P., Kuchař, A., Jacobi, C., and Pišoft, P.: Enhanced internal gravity wave activity and breaking over the northeastern Pacific–eastern Asian region, Atmos. Chem. Phys., 15, 13097–13112, https://doi.org/10.5194/acp-15-13097-2015, 2015.

---

## Referee Comment (RC1) · Anonymous Referee #2 · 14 May 2020

General comment:

The study is well organized and well written about a relatively new topic which is of high interest. Thus, I recommend a publication in AMT after a minor revision. I have two points which need a further discussion by the authors.

1) Why do the authors often use TIMED/SABER measurements in their studies which have a poor horizontal resolution? For example, the horizontal sampling of Aura/MLS is better. It would be important to know why the approach of Jiang and Wu disappeared after 2005 or so. They used high resolution brightness temperature measurements of MLS. It makes more sense to discover gravity waves in satellite data sets having a high

horizontal resolution.

2) line 349 "... which indicates real gravity wave activity" This seems to be a speculation since there are steep meridional and zonal gradients at the polar vortex edge which might be misinterpreted as enhanced gravity wave activity around the polar vortex. Please discuss this fundamental problem in more detail.

————————————————————

---

## Referee Comment (RC2) · Anonymous Referee #1 · 19 May 2020

This article presents an organised, informative, and well written study on the important topic of separating inertial instability signals from gravity waves. I recommend this study for publication in AMT after minor revisions.

General Comments: Please go through the text, captions, and figure axis labels including the word 'perturbation' and consider replacing the word or adding language to be more explicit. At different times in the article 'perturbation' can refer to difference, standard deviation, or variance. I found myself as a reader having to go back in the text and remind myself what type of perturbation was being discussed.

Specific Comments: P3L130-135: Uncertainty estimates of pointing jitter found in

[Figure]

Remsberg et al. (2008) may be underestimated. A French lidar study Wing et al. (2018) showed larger than expected variations in SABER temperatures when compared to co-located lidar at OHP. Variations were partially attributed to pointing jitter. The same study also reported a seasonally evolving geopotential bias which had a complex altitude dependence. It might be worth mentioning in Section 4.1 the minimum precision of SABER temperatures may not account for any additional uncertainty coming from inaccuracy in the altitude.

P7L192: Please add a brief justification for the critical wavelength of 15 km. Maybe introduce some of the information from P11L314 here or indicate that the choice will be discussed later. It was not perfectly clear on the first read through.

P14L413: please add the word variance somewhere in this sentence for clarity

Technical Corrections: P10L276: "squared"

Figure 1 Caption "positive"

---

## Author Comment (AC1) · 12 Jul 2020

Dear Dr. Šácha,

thank you very much for your valuable comments on our article. We have corrected our mistake in citation and addressed your specific questions in more detail below.

Yours sincerely,

Cornelia Strube

[Figure]

- *[...]I just noticed that on P3L63 you reference Šácha et al. (2016), but it should be either Šácha et al. (2015) or Šácha et al. (2014), which are observational studies.[...]*

  Sorry for the wrong citation, we meant to refer to Šácha et al. (2014) here. We have adjusted this and added the second citation as well.

- *1) When using some simple form of vertical background separation (e.g. polynomial fit) is it possible to identify the potential presence of the inertial instability signal in the data from analysis of the vertical wavenumber power spectrum density of disturbances? I.e.as long as the spectrum slope follows the theoretical saturated GW spectrum, then weare on the safe side?*

  The critical vertical wavenumber separating the saturated and unsaturated part of the vertical wavenumber spectrum is rather short in the stratosphere, corresponding to about 3 km vertical wavelength (Smith et al., 1987; Dewan et al., 1984; Thomas et al., 1992; Warner and McIntyre, 2001). Inside this range, inertial instabilities would also become convectively instable with the same threshold. This would not allow for a safe separation. In addition, for conveying momentum in the middle atmosphere we are also interested in GWs of longer vertical wavelengths not yet saturated. Therefore one would rather have to search for additional different criteria.

- *2) As you acknowledge in the paper, all GW types are highly sensitive to the background winds during their lifecycle, can you exclude the possibility that a portion of GW activity can be filtered out using the horizontal filtering? How do you expect GWs to interact with the inertial instability region?*

  Horizontal filtering is likely to remove some very long horizontal wavelength GWs. However, since GWMF is inversely proportional to the horizontal wavelength the meso- and short-scale GWs are of larger importance for many analyses. In addition, wind modulation implies stronger effects on the vertical wavelengths than

the horizontal wavelengths. Thus, for the purpose of following the same wave through different altitudes, horizontal filtering should be preferred to vertical filtering, as it will usually not shift waves in and out of an observational filter depending on the altitude.

**References**

Šácha, P. and Foelsche, U. and Pioft, P.: *Analysis of internal gravity waves with GPS RO density profiles*, Atmos. Meas. Tech., 7, 4123–4132, https://doi.org/10.5194/amt-7-4123-2014, 2014.

Šácha, P., Kuchař, A., Jacobi, C., and Pišoft, P.: *Enhanced internal gravity wave activity and breaking over the northeastern Pacific–eastern Asian region*, Atmos. Chem. Phys., 15, 13097–13112, https://doi.org/10.5194/acp-15-13097-2015, 2015.

Smith, S. A., D. C. Fritts, and T. E. Vanzandt, 1987: *Evidence for a Saturated Spectrum of Atmospheric Gravity Waves.* J. Atmos. Sci., 44, 1404–1410, https://doi.org/10.1175/1520-0469(1987)044<1404:EFASSO>2.0.CO;2.

Dewan, E.M., Grossbard, N., Quesada, A.F. and Good, R.E. (1984), *Spectral analysis of 10m resolution scalar velocity profiles in the stratosphere*. Geophys. Res. Lett., 11: 80-83. doi:10.1029/GL011i001p00080

Thomas, L., I. T. Prichard, and I. Astin: *Radar observations of aninertia-gravity wave in the troposphere and lower stratosphere*,Ann. Geophysicae,10,690-697, 1992.

Warner, C. D., and M. E. McIntyre, 2001: *An Ultrasimple Spectral Parameterization for Nonorographic Gravity Waves.* J. Atmos. Sci., 58, 1837–1857, https://doi.org/10.1175/1520-0469(2001)058<1837:AUSPFN>2.0.CO;2.
* * *

---

## Author Comment (AC2) · 12 Jul 2020

Dear Anonymous Referee #2,

thank you very much for your valuable comments on our article. We have addressed your two specific questions in more detail below and revised the manuscript accordingly.

Yours sincerely,
Cornelia Strube

[Figure]

*1) Why do the authors often use TIMED/SABER measurements in their studies which have a poor horizontal resolution? For example, the horizontal sampling of Aura/MLS is better. It would be important to know why the approach of Jiang and Wu disappeared after 2005 or so. They used high resolution brightness temperature measurements of MLS. It makes more sense to discover gravity waves in satellite data sets having a high horizontal resolution.*

Gravity wave variances calculated from MLS measurements using sublimb geometry were used also in a few studies later than 2005, see for example in Wu and Eckermann (2008). Still, we agree with Referee #2, it is a pity that this dataset is not used more extensively in newer studies. However, we focus here on studies of vertical measurement profiles which can be used to infer GWMF as well. Therefore, including MLS observations is beyond the scope of our study. We included a short comments on the matter in the introduction (see L67 and L89).

*2) line 349 "... which indicates real gravity wave activity" This seems to be a speculation since there are steep meridional and zonal gradients at the polar vortex edge which might be misinterpreted as enhanced gravity wave activity around the polar vortex. Please discuss this fundamental problem in more detail.*

Of course, steep temperature gradients occur at the vortex edge and can bias estimates of GW activity. However, the feature observed here is relatively broad and extends with similar strength over almost 30° of latitude. In particular, it is also observed in the jet core and thus away from particular strong horizontal gradients. We therefore think that this is a robust feature. We add the following sentence in the manuscript (L349):

... and in particular in a wide range of latitudes associated with the polar jet. Though there may be some structures caused by horizontal gradients at the vortex edge, this enhancement spans approx. 30° of latitude and hence indicates real gravity wave activity.

**References**

Wu, D. L., and S. D. Eckermann, 2008: *Global Gravity Wave Variances from Aura MLS: Characteristics and Interpretation.* J. Atmos. Sci., 65, 3695–3718, https://doi.org/10.1175/2008JAS2489.1.

---

## Author Comment (AC3) · 12 Jul 2020

Dear Anonymous Referee #1,

thank you very much for your valuable comments on our article. Please find a detailed response to your specific questions below. We have also revised the manuscript to make the use of the term "perturbation" more clear as well as to address your technical corrections.

Yours sincerely,
Cornelia Strube

[Figure]

- *Please go through the text, captions, and figure axis labels including the word 'perturbation' and consider replacing the word or adding language to be more explicit. At different times in the article 'perturbation' can refer to difference, standard deviation, or variance. I found myself as a reader having to go back in the textand remind myself what type of perturbation was being discussed.*

  We have revised the text, captions and figure labels for the use of the term "perturbation". In particular, we have specified the term by adding a "gravity wave" in front of the "temperature perturbation" for the instances where we refer to calculated temperature perturbations according to any of the background removal methods introduced in Section 3. This indicates that the resulting perturbations are expected to be mainly comprised of gravity wave induced perturbations (disregarding spurious remnants from other processes like inertial instabilities). Furthermore, we have changed the colorbar labels in Fig. 6 and 7 into "Mean squared temperature perturbations" instead of "Temperature variances", which was a genuine mistake in labeling. Please see the supplement with highlighted changes to find the specific instances.

- *P3L130-135: Uncertainty estimates of pointing jitter found in Remsberg et al. (2008) may be underestimated. A French lidar study by Wing et al. (2018) showed larger than expected variations in SABER temperatures when compared to co-located lidar at OHP. Variations were partially attributed to pointing jitter. The same study also reported a seasonally evolving geopotential bias which had acomplex altitude dependence. It might be worth mentioning in Section 4.1 the minimum precision of SABER temperatures may not account for any additional uncertaintycoming from inaccuracy in the altitude.*

  We added the following sentences on p5, l132:
  Somewhat stronger pointing jitter than mentioned in Remsberg et al. (2008) was recently reported by Wing et al. (2018). Further, there is a geopotential bias with complex vertical structure. Therefore precision estimates in Remsberg et al.

(2008) may be somewhat too low.

- *P7L192: Please add a brief justification for the critical wavelength of 15 km. Maybe introduce some of the information from P11L314 here or indicate that the choice will be discussed later. It was not perfectly clear on the first read through.*

  For our study, we chose the cutoff vertical wavelength for the Butterworth filter according to the suggestion from previous studies (Ehard et al., 2015; Ehard et al., 2017; Rapp et al., 2018). Ehard et al. (2015) established the advantages of using a Butterworth filter analysis for single vertical lidar profiles over several other methods pointing out especially the applicability of the Butterworth filter for a broad passband and the adjustability in the cutoff wavelength. Ehard et al. (2017) explained the use of a 15km cutoff wavelength to "limit the contribution of the stratopause to the gravity wave temperature perturbations" which in our datasets is of less relevance. Rapp et al. (2018) transferred the use of the filter to single satellite profiles and expected the specific separation to "work well except for in the tropical stratosphere, where Kelvin waves are known to occur with vertical wavelengths well below 15km". We use it as a reference here for vertical filtering as it was applied in other studies.

- *P14L413: please add the word variance somewhere in this sentence for clarity*

  We have replaced the word perturbation with variance in this line.

- *Technical Corrections: P10L276: "squared"*

- *Figure 1 Caption "positive"*

  We have corrected the misspellings in L276 and the caption of Figure 1.
**References**

Ehard, B., Kaifler, B., Kaifler, N., and Rapp, M.: *Evaluation of methods for gravity wave extraction from middle-atmospheric lidar temperature measurements*, Atmos. Meas. Tech., 8, 4645–4655, https://doi.org/10.5194/amt-8-4645-2015, 2015.

Ehard, B., et al. (2017): *Horizontal propagation of large‐amplitude mountain waves into the polar night jet*, J. Geophys. Res. Atmos., 122, 1423– 1436, doi:10.1002/2016JD025621.

Rapp, M., Dörnbrack, A., and Kaifler, B.: *An intercomparison of stratospheric gravity wave potential energy densities from METOP GPS radio occultation measurements and ECMWF model data*, Atmos. Meas. Tech., 11, 1031–1048, https://doi.org/10.5194/amt-11-1031-2018, 2018.

---

## Author Response (AR1)

**Authors' response for open-discussion comments on "Removing spurious inertial instability signals from gravity wave temperature perturbations using spectral filtering methods"**

Cornelia Strube[1], Manfred Ern[1], Peter Preusse[1], and Martin Riese[1]

[1]Institut für Energie- und Klimaforschung – Stratosphäre (IEK–7), Forschungszentrum Jülich GmbH, 52425 Jülich, Germany

**Correspondence:** Cornelia Strube (c.strube@fz-juelich.de)

First of all, we would like to thank the two anonymous referees and Petr Šácha for their valuable comments. We have addressed all of them in more detail below (Comments in italic and corresponding answers in regular font). The manuscript was adjusted accordingly. Below, we also enclose a version of the manuscript highlighting all changes in the text.

**Response to Comments by Anonymous Referee #1**

5   – *Please go through the text, captions, and figure axis labels including the word 'perturbation' and consider replacing the word or adding language to be more explicit. At different times in the article 'perturbation' can refer to difference, standard deviation, or variance. I found myself as a reader having to go back in the text and remind myself what type of perturbation was being discussed.*

We have revised the text, captions and figure labels for the use of the term "perturbation". In particular, we have speci-
10   fied the term by adding a "gravity wave" in front of the "temperature perturbation" for the instances where we refer to calculated temperature perturbations according to any of the background removal methods introduced in Section 3. This indicates that the resulting perturbations are expected to be mainly comprised of gravity wave induced perturbations (disregarding spurious remnants from other processes like inertial instabilities). Furthermore, we have changed the colorbar labels in Fig. 6 and 7 into "Mean squared temperature perturbations" instead of "Temperature variances", which was a
15   genuine mistake in labeling. Please see the supplement with highlighted changes to find the specific instances.

  – *P3L130-135: Uncertainty estimates of pointing jitter found in Remsberg et al. (2008) may be underestimated. A French lidar study by Wing et al. (2018) showed larger than expected variations in SABER temperatures when compared to co-located lidar at OHP. Variations were partially attributed to pointing jitter. The same study also reported a seasonally evolving geopotential bias which had a complex altitude dependence. It might be worth mentioning in Section 4.1 the*
20   *minimum precision of SABER temperatures may not account for any additional uncertainty coming from inaccuracy in the altitude.*

We added the following sentences on p5, l132:
Somewhat stronger pointing jitter than mentioned in Remsberg et al. (2008) was recently reported by Wing et al. (2018).

Further, there is a geopotential bias with complex vertical structure. Therefore precision estimates in Remsberg et al. (2008) may be somewhat too low.

- *P7L192: Please add a brief justification for the critical wavelength of 15 km. Maybe introduce some of the information from P11L314 here or indicate that the choice will be discussed later. It was not perfectly clear on the first read through.*

  For our study, we chose the cutoff vertical wavelength for the Butterworth filter according to the suggestion from previous studies (Ehard et al., 2015, 2017; Rapp et al., 2018). Ehard et al. (2015) established the advantages of using a Butterworth filter analysis for single vertical lidar profiles over several other methods pointing out especially the applicability of the Butterworth filter for a broad passband and the adjustability in the cutoff wavelength. Ehard et al. (2017) explained the use of a 15 km cutoff wavelength to "limit the contribution of the stratopause to the gravity wave temperature perturbations" which in our datasets is of less relevance. Rapp et al. (2018) transferred the use of the filter to single satellite profiles and expected the specific separation to "work well except for in the tropical stratosphere, where Kelvin waves are known to occur with vertical wavelengths well below 15 km". We use it as a reference here for vertical filtering as it was applied in other studies.

- *P14L413: please add the word variance somewhere in this sentence for clarity*

  We have replaced the word perturbation with variance in this line.

- *Technical Corrections: P10L276: "squared"*

- *Figure 1 Caption "positive"*

  We have corrected the misspellings in L276 and the caption of Figure 1.

**Response to Comments by Anonymous Referee #2**

*1) Why do the authors often use TIMED/SABER measurements in their studies which have a poor horizontal resolution? For example, the horizontal sampling of Aura/MLS is better. It would be important to know why the approach of Jiang and Wu disappeared after 2005 or so. They used high resolution brightness temperature measurements of MLS. It makes more sense to discover gravity waves in satellite data sets having a high horizontal resolution.*

Gravity wave variances calculated from MLS measurements using sublimb geometry were used also in a few studies later than 2005, see for example in Wu and Eckermann (2008). Still, we agree with Referee #2, it is a pity that this dataset is not used more extensively in newer studies. However, we focus here on studies of vertical measurement profiles which can be used to infer GWMF as well. Therefore, including MLS observations is beyond the scope of our study. We included a short comments on the matter in the introduction (see L67 and L89).

*2) line 349 "... which indicates real gravity wave activity" This seems to be a speculation since there are steep meridional and zonal gradients at the polar vortex edge which might be misinterpreted as enhanced gravity wave activity around the polar vortex. Please discuss this fundamental problem in more detail.*

55     Of course, steep temperature gradients occur at the vortex edge and can bias estimates of GW activity. However, the feature observed here is relatively broad and extends with similar strength over almost 30° of latitude. In particular, it is also observed in the jet core and thus away from particular strong horizontal gradients. We therefore think that this is a robust feature. We add the following sentence in the manuscript (L349):

    ... and in particular in a wide range of latitudes associated with the polar jet. Though there may be some structures caused by

60 horizontal gradients at the vortex edge, this enhancement spans approx. 30° of latitude and hence indicates real gravity wave activity.

**Response to Short comment by Petr Šácha**

- *[...]I just noticed that on P3L63 you reference Šácha et al. (2016), but it should be either Šácha et al. (2015) or Šácha et al. (2014), which are observational studies.[...]*

65     Sorry for the wrong citation, we meant to refer to Šácha et al. (2014) here. We have adjusted this and added the second citation as well.

- *1) When using some simple form of vertical background separation (e.g. polynomial fit) is it possible to identify the potential presence of the inertial instability signal in the data from analysis of the vertical wavenumber power spectrum density of disturbances? I.e.as long as the spectrum slope follows the theoretical saturated GW spectrum, then weare on*

70     *the safe side?*

    The critical vertical wavenumber separating the saturated and unsaturated part of the vertical wavenumber spectrum is rather short in the stratosphere, corresponding to about 3 km vertical wavelength (Smith et al., 1987; Dewan et al., 1984; Thomas et al., 1992; Warner and McIntyre, 2001). Inside this range, inertial instabilities would also become convectively instable with the same threshold. This would not allow for a safe separation. In addition, for conveying momentum in the

75 middle atmosphere we are also interested in GWs of longer vertical wavelengths not yet saturated. Therefore one would rather have to search for additional different criteria.

- *2) As you acknowledge in the paper, all GW types are highly sensitive to the background winds during their lifecycle, can you exclude the possibility that a portion of GW activity can be filtered out using the horizontal filtering? How do you expect GWs to interact with the inertial instability region?*

80     Horizontal filtering is likely to remove some very long horizontal wavelength GWs. However, since GWMF is inversely proportional to the horizontal wavelength the meso- and short-scale GWs are of larger importance for many analyses. In addition, wind modulation implies stronger effects on the vertical wavelengths than the horizontal wavelengths. Thus, for the purpose of following the same wave through different altitudes, horizontal filtering should be preferred to vertical filtering, as it will usually not shift waves in and out of an observational filter depending on the altitude.

[revised manuscript text omitted]